# Double-Facet Effect of Artificial Mechanical Stress on Red Blood Cell Deformability: Implications for Blood Salvage

Tamir Tsohar [1], Shaul Beyth [1], Alexander Gural [2], Dan Arbell [3], Saul Yedgar [4] and Gregory Barshtein [4,*]

[1] Orthopedic Department, Hadassah-Hebrew University Medical Center, Jerusalem 9112001, Israel
[2] Blood Bank, Hadassah-Hebrew University Medical Center, Jerusalem 9112001, Israel
[3] Department of Pediatric Surgery, Hadassah-Hebrew University Medical Center, Jerusalem 9112001, Israel
[4] Department of Biochemistry, Faculty of Medicine, Campus Ein Kerem, Hebrew University, Jerusalem 9190501, Israel
[*] Correspondence: gregoryba@ekmd.huji.ac.il; Tel.: +972-2675-8309

**Abstract:** The use of intra-operative blood salvage, dialysis, and artificial organs are associated with the application of non-physiological mechanical stress on red blood cells (RBCs). To explore the effect of these procedures on red cell deformability, we determined it before and after the mechanical stress application both in an in vitro system and following a blood-saving procedure. RBC from eight healthy donors and fifteen packed RBC units were subjected to mechanical stress. RBCs from five patients undergoing orthopedic surgery were also collected. We measured the percent of undeformable cells (%UDFC) in the red cell samples using our cell flow properties image analyzer, which provides the distribution of RBC deformability in a large cell population. Mechanical stress systematically reduced the cell deformability and increased the %UDFC, while simultaneously causing hemolysis of rigid, undeformable RBCs. Ultimately, the overall result depended on the initial level of the undeformable cells; the stress-induced change in the proportion of rigid cells (Δ%UDFC) increased (Δ%UDFC > 0) when its initial value was low, and decreased (Δ%UDFC < 0) when its initial value was high. This suggests that the final impact of mechanical stress on the percent of rigid cells in the RBC population is primarily determined by their initial concentration in the sample.

**Keywords:** red blood cells; deformability; shear stress; mechanical stress; blood salvage

## 1. Introduction

Significant blood loss during surgery may be acceptable to a certain degree [1,2]. However, a rapid and significant decrease in blood volume may lead to hypovolemic shock [3,4]. In addition, the red blood cell (RBC) loss harms organ vitality [5,6].

To maintain blood volume in a bleeding patient, the currently available options include the transfusion of stored, packed RBC (PRBC) from an autologous [7] or allogeneic source [8] and blood salvage [9] throughout the surgery. These techniques are not faultless, and their effects on RBCs can be considerable. RBCs are constantly subjected to shear stress in circulation, affecting their functionality [10–14]. An even more significant effect on RBC physiological properties is exerted by flow through various types of artificial devices [10,15–17] or by blood processing procedures (e.g., dialysis, salvage) [18–20], where the level of shear stress can be significantly higher than physiological, as reported in numerous studies [16,19,21,22].

Several strategies can be used to diminish the transfusion of allogeneic RBCs; blood salvage is one of them [9]. Blood-saving throughout surgery is becoming increasingly popular as a strategy to reduce allogeneic blood transfusions. This approach eliminates the risk of immunological transfusion reactions and infectious disease transmission [9]. It is a cost-effective procedure to avoid RBC loss, reducing the need for allogeneic red cell transfusions [23,24]. Many devices are available for blood salvage during surgery [23,24].

Nevertheless, some studies have reported significant variability in the quality of the obtained RBCs with a significant elevation in plasma-free hemoglobin in the circulation

of patients undergoing perioperative auto-transfusion [18,25–27]. However, since during the RBC collection and purification, they are exposed to mechanical stress, blood salvage is associated with a reduction in the lifespan of erythrocytes [28–30]. Although the effect of mechanical action is beyond doubt, the detailed mechanism of the influence of blood salvage procedure on the functionality of RBCs remains under discussion [29].

Depending on the level of shear stress and its duration, the exposure can lead to cell hemolysis. [31–37]. However, even before cell destruction, there is a change in the properties of the cytosol, cytoskeleton, and cell membrane [12]. The RBC undergoes structural changes in response to the supraphysiological mechanical stress [16,36,38–40]. Prolonged exposure to supraphysiological shear stress may promote the spread of abnormal erythrocyte morphology [22], phosphatidylserine externalization [41], and increased vesiculation [41,42]. Ultimately, these changes lead to the destruction of RBCs [31–37]. The effect of mechanical stress on erythrocytes was analyzed using a numerical approach [35], in vitro and ex vivo experiments [32,33,36,37,43,44].

Freitas Leal et al. [41] demonstrated that the flow of stored RBCs through a heart-lung machine provoked alterations in RBC structure and functionality. These include elevation in osmotic fragility, alteration in cell aggregability and deformability, and acceleration of microvesicle formation. The degree and kinetics of these changes depend on the cell storage duration in the blood bank. The article's authors [41] conclude that their data will help develop, improve, and control the quality of the extracorporeal circuit approach.

Watanabe et al. [14] used a shear (counter-rotation) system coupled to a microscope to directly monitor RBC fragmentation and hemolysis, which were recognized after exposure to shear stress of 288 Pa for 40 s. Later, the same research group [12] visualized the presence of abnormal RBCs (with damaged morphology) under prolonged (more than 100 s) exposure to shear stress of 60 Pa that induced asymmetric cell elongation. In regards to physiological conditions, Zhu et al. [45] demonstrated (using model simulations of the flow of erythrocytes through the human spleen's venous fissures) that erythrocytes' deformation can contribute to cell membrane vesiculation and subsequent decrease in its surface-area-to-volume ratio. These changes should lead to an alteration in the cell's functionality, for example, an increase in its rigidity [46,47].

Deformability is one of the vital features of RBCs. Under circulating conditions, healthy erythrocytes are easily deformed, facilitating their passage through narrower microvessels. RBCs with increased rigidity weaken perfusion and oxygen delivery to tissues [48–51] and can block microvessels [51–53]. Relatively rigid (e.g., aged) RBCs prevent the passage of the cells in the spleen vasculature and increase splenic RBC sequestration and destruction [54–56]. Previously, we demonstrated that the transfusion of stored units with a high fraction of rigid (undeformable) RBCs provoke impaired blood skin perfusion [57]. Therefore, the alteration in cell deformability induced by mechanical stress (for example, during perioperative salvage [18]) is highly significant.

In a previous study [58], we examined how cell deformability is altered during the preparation of donated blood for storage as a packed RBC (PRBC) unit. This procedure, which includes centrifugation and filtration, affects cell membrane composition, leading to mixed results [56,59,60]. We have previously shown [58] that, depending on the initial level of the donor RBCs' deformability, both a decrease and an increase in the fraction of undeformable cells can occur. The change in the overall deformability depended on the initial level of undeformable cells (%UDFC); it increased when the baseline %UDFC was low and decreased when it was high. Based on our data, we hypothesized that exposing cells to high shear stress during the preparation of a PRBC unit, is likely to have two opposite effects: first, the destruction of undeformable RBCs, thereby reducing their percent in the population; second, mechanical damage to the RBC membrane with a subsequent increase in the cell's rigidity (thereby increasing the %UDFC). Consequently, the final effect of the process of unit preparation is determined primarily by the initial concentration of undeformable cells in the blood collected from the donor.

To substantiate this hypothesis, we examined the shear stress effect on RBC deformability using two other methods of shear stress application: (1) in vitro application of mechanical stress to RBCs and (2) blood salvage procedures applied in the operating rooms for auto-transfusion of patients' blood.

## 2. Materials and Methods

Experimental design: Mechanical stress was applied to RBCs using two methods (see Figure 1):

1. RBCs suspended in PBS buffer were rolled with steel beads [61].
2. Blood salvage procedure, in which the whole blood of a patient undergoing orthopedic surgery was collected and processed for re-administration to the patient, using the OrthoPAT (Haemodynamics, Boston, MA, USA). RBCs were collected before and after the mechanical stress exposure, and cell deformability was assessed as described below.

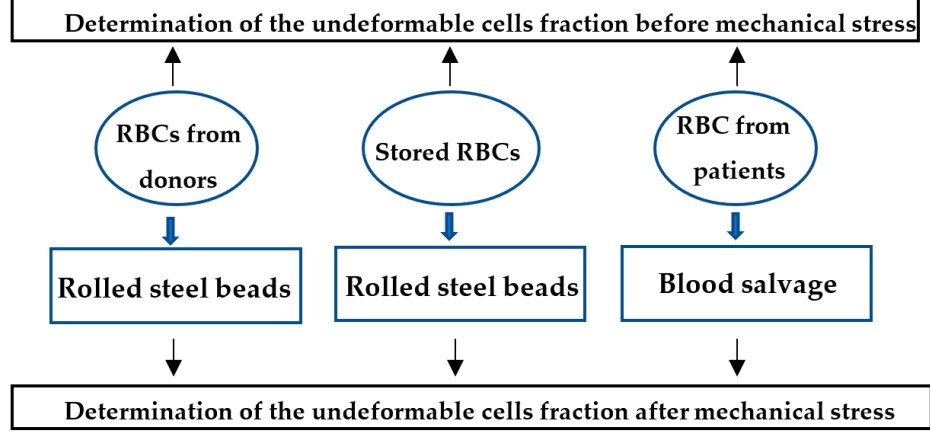

**Figure 1.** Flowchart of the experimental design.

### 2.1. Materials

Antiaerosol pipette tips (1 mL, catalog T1000.96) were purchased from Neptune Scientific (San Diego, CA, USA). Stainless steel beads (catalog Kit8589) were purchased from Nation-Skander California Co. (Anaheim, CA, USA). Bovine Serum Albumin (catalog A3294) was purchased from Sigma (St. Louis, MO, USA); phosphate-buffered saline (PBS) without calcium and magnesium (catalog 02-023-1A) was purchased from B.I. (Kibbutz Beit-Haemek, Israel).

### 2.2. Freshly Donated Blood

Blood samples were collected from eight healthy volunteers after obtaining an Institutional Helsinki Committee Regulations Permit (98290, Hadassah Hospital, Jerusalem, Israel). The blood was collected into vacuette tubes (Greiner Bio-One, Frickenhausen, Austria) containing K3EDTA.

### 2.3. RBC from Packed RBC Units (PRBC)

Blood was drawn from fifteen Hadassah Hospital Blood Bank donors into standard sterile bags containing citrate-phosphate-dextrose (Fresenius Kabi AG, Homburg, Germany). Collected blood was centrifuged (Roto Silenta 630 RS, Tuttlingen, Germany) for 6 min ($1754\times$ *g*, at room temperature) for RBC isolation. Units of concentrated RBCs (non-leukofiltrated) were stored in CPDA-1 under 2–6 °C. In the presents study we used outdated packed RBC units with 35–38 day storage duration. These cells are stored (under standard conditions) for a maximum of three days after the unit expires. Five ml were drawn from the PRBC unit and the RBCs were washed in PBS (twice) by centrifugation ($500\times$ *g* for 10 min).

### 2.4. Preparation of RBC from Freshly Collected Blood

RBCs were isolated from freshly collected blood by centrifugation, and washed three times by centrifugation ($500\times g$ for 10 min) in PBS.

### 2.5. Application of Mechanical Stress

Mechanical stress was induced using the previously described setup [61,62], which is widely used to characterize cells' mechanical fragility [10,11,33,63–69]. In brief, 3 mL of RBC (10%) suspension in PBS were rocked (at 40 cycles/min) for one hour at room temperature in glass test tubes (13 mm $\times$ 100 mm) containing five steel beads (3.175 mm). This conventional method leads to shear stress-induced hemolysis of less than 5% [61,62] and is thus considered moderate mechanical stress. After the treatment, RBCs were isolated and washed once in PBS and then resuspended in PBS supplemented with 0.5% albumin for deformability measurements.

### 2.6. Surgery Patients

In an Institutional Review Board-approved cohort study, five patients (four females, one male, ages 14–24) scheduled for elective multiple-level spine fusion, underwent pre-surgery workouts, including blood count and anesthesiologist examination.

### 2.7. RBC from Salvage Procedure

RBCs were twice washed and resuspended in $Ca^{2+}/Mg^{2+}$-phosphate-buffered saline (PBS), pH = 7.4. The OrthoPAT (orthopedic perioperative auto-transfusion system) process is divided into the collection, filtering, washing, and re-infusion phases. RBCs from the operative field are collected using a dedicated double-lumen suction device. One lumen suctions blood from the operative field, and the other lumen adds a predefined portion of heparinized buffer to the salvaged blood. The collected blood is then filtrated and collected into a reservoir. Component separation is reached by a precision centrifuge. RBCs are then washed and filtered through a semi-permeable membrane, and salvaged RBCs are resuspended in the buffer (Hct = 50–80%). The obtained cells may subsequently be transfused back into the patient at any time within a six-hour window.

### 2.8. Preparation of RBC from the Operating Room

RBCs were collected from the patient's blood (before the operation and after RBC salvaging by OrthoPAT), washed (in PBS) by centrifugation, and then resuspended in the phosphate-buffered saline supplemented with 0.5% of albumin for the following measurements of cell deformability.

### 2.9. RBC Deformability

RBC deformability was determined using our original cell flow properties analyzer (CFA), where the change of RBC shape was directly visualized in microfluidic (with a 200 µm gap) under a shear stress of 3.0 Pa [46,58,62]. 50 µL of RBC suspension (Hct = 1%, phosphate-buffered saline supplemented with 0.5% of albumin) are inserted into the microfluidic and incubated for 15 min (at room temperature) when the RBC are attached to the glass slide. The attached cells in the microfluidic are then subjected to a buffer flow, and the deformability of cells is determined at a shear stress of 3.0 Pa. During the measurements, 14–21 images of randomly chosen fields (with an area of 0.1 mm$^2$) are collected [46,58,62]. RBC deformability is determined by the shear stress-induced elongation and expressed by the elongation ratio (ER), namely, the ratio between the major (a) and minor (b) axes, according to the equation ER = a/b, where ER = 1 reflects a spherical RBC that is not deformed at the shear stress applied. This procedure is applied to each RBC in a population of 8000–10,000 cells and provides the deformability distribution in the cell population, which allows for the derivation of various deformability parameters [46,58,62]. In the present study, we focused on the percentage of undeformable cells (%UDFC) in the cell population characterized by ER $\leq$ 1.1. Our choice is explained by the fact that this

parameter characterizes the size of the fraction of rigid cells capable of hindering adequate blood microcirculation.

The percent of UDFC was derived for each RBC sample (freshly donated, stored, and salvaged during surgery) before and after the application of mechanical stress (by rolling with steel beads or salvage processing). We calculated the %UDFC change (Δ%UDFC) from the difference in the percentage of undeformable cells before and after the application of mechanical stress (by rolling steel beads or blood salvage).

### 2.10. Statistical Analysis

All data are presented as mean ± SD. We examined statistical significance using the paired two-tailed Student's *t*-test (SPSS 21). *p* values were considered significant at $p < 0.05$.

## 3. Results

### 3.1. Effect of Moderate Mechanical Stress on RBC Deformability

Figure 2 shows the impact of moderate mechanical stress (hemolysis < 5%) on the portion of rigid, undeformable cells (%UDFC) in both freshly-collected and cold-stored RBC. This figure demonstrated that the application of shear stress led to both an increase and a decrease in %UDFC. Furthermore, we did not observe a significant difference in the concentration of undeformable cells between samples obtained before and after the exposure to mechanical stress. Thus, the median value of %UDFC was 2.63% for cells before (RBC$_B$) treatment and 3.68% for mechanically-treated cells (RBC$_{MS}$), *p* = 0.45.

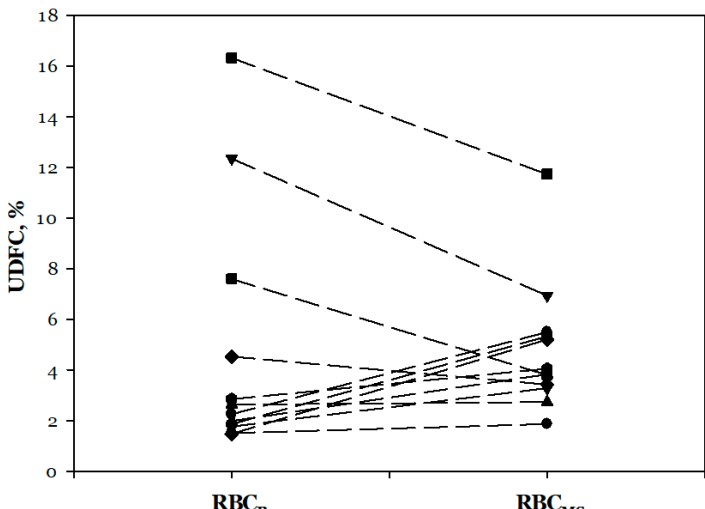

**Figure 2.** Percentage of undeformable cells (%UDFC) in samples collected before (RBC$_B$) and after mechanical stress application (RBC$_{MS}$). Mechanical stress was applied to cells by rolling freshly donated and stored RBCs with steel beads. Identical symbols designate the same RBC sample before and after applying mechanical stress. No significant difference between the two groups was observed, *p* = 0.45.

In addition, we analyzed the impact of mechanical stress separately for freshly donated and stored RBCs. Figure 3 shows that the change in the proportion of rigid cells is primarily positive (ΔUDFC > 0) for freshly-donated cells and negative for stored RBC. Thus, mechanical stress leads predominantly to an increase in the fraction of non-deformable cells in the case of freshly donated RBCs, and, conversely, to a decrease in the fraction of rigid cells when stored RBCs are analyzed. The observed difference in the response to mechanical stress seems to be related to the fact that the initial concentration of undeformable cells is significantly higher for samples of stored RBCs than fresh ones. Furthermore, as we have previously shown [61], these are the rigid cells that are predominately destroyed as a result of mechanical impact. Therefore, the %UDFC predominately decreases when stored cells are exposed to mechanical stress.

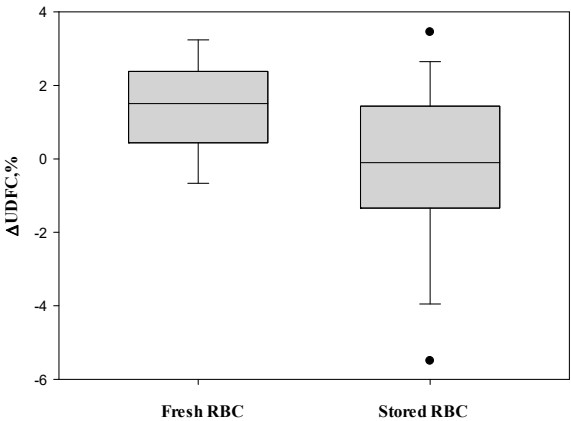

**Figure 3.** Alteration (after rolling cells with steel beads) of the percentage of undeformable cells (ΔUDFC, %) in the RBC population for fresh and stored samples. No significant difference between the two groups, *p* = 0.154.

Based on the results presented in Figure 2, we divided all the samples (whether fresh or stored) into two groups based on the percent of non-deformable RBCs before mechanical stress application. As a cutoff parameter, we chose the median value of the percentage of undeformable cells in the sample (2.63%, as mentioned earlier). As demonstrated in Figure 4, for both fresh and stored RBCs, the application of mechanical stress decreased the level of rigid cells (ΔUDFC < 0) when its initial value (before treatment) was higher than the median (2.63%), and increased it (ΔUDFC > 0) when the initial value was lower than the median.

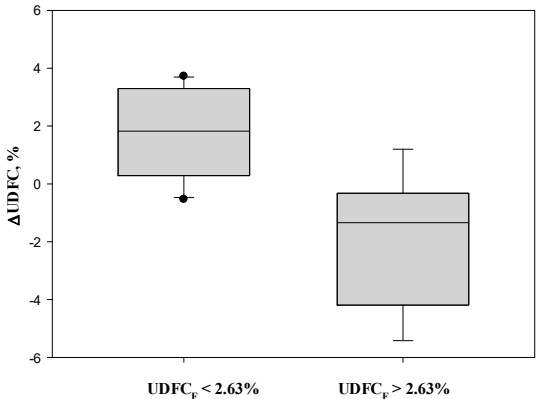

**Figure 4.** Alteration in the percentage of undeformable cells (ΔUDFC, %) in the RBC population following mechanical stress (applied to cells by rolling with steel beads). *p* = 0.0007.

Thus, we found that samples with a low concentration of undeformable cells subjected to mechanical stress predominately showed an increase in this fraction, while samples with a high concentration of such cells (predominantly stored cells) mostly showed a decrease in this fraction.

Furthermore, (Figure 5) the mechanical stress-induced change in the level of rigid cells (ΔUDFC,%) showed a significant linear correlation (r = 0.87; *p* = 1.4 × 10⁻⁷) with the proportion of the rigid cells before treatment (%UDFC$_B$).

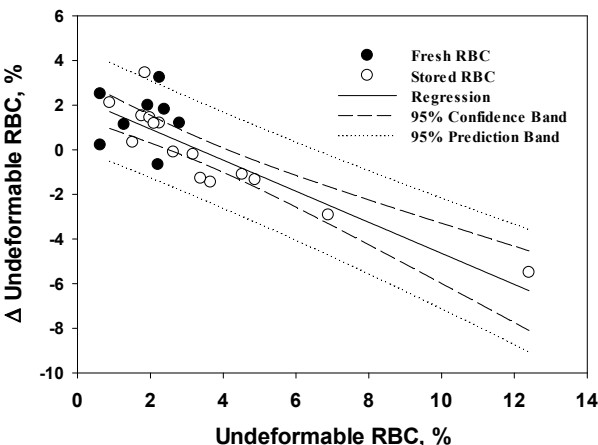

**Figure 5.** Relationship between ΔUDFC and %UDFC$_B$ described by linear regression (r = 0.87; *p* = 1.4 × 10$^{-7}$).

### 3.2. Effect of Blood Salvage on RBC Deformability

Figure 6 depicts representative images of a patient's RBCs before and after the application of cell salvage (Figure 6A) and the average distribution curves of RBC deformability, expressed by cell elongation ratio (ER), for the five surgery patients (Figure 6B). This figure shows that, on the average, the ER distribution curve does not change as a result of the salvage procedure.

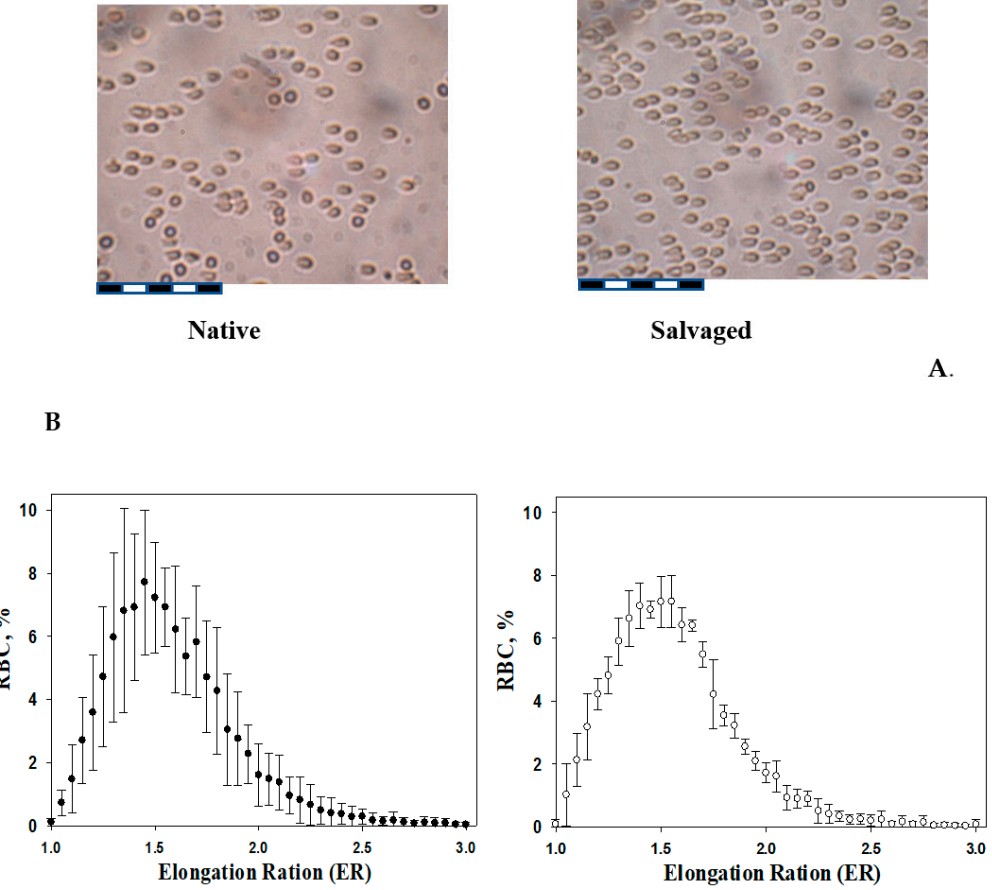

**Figure 6.** Deformability of freshly collected and salved RBCs. (**A**). Representative images of deformed RBCs for freshly collected and salved RBCs. Scale bars are 50 μm (5 μm × 10 μm). (**B**). Generalized Elongation Ratio (ER) distribution curves calculated for five paired samples collected before (left panel) and after (right panel) salvage.

The results are summarized in Table 1. The data show that blood salvage does not change the average number of undeformable cells (obtained for five blood samples), similar to the effect of rolling the cells with metal balls, depicted above (see Figure 2).

**Table 1.** Characterizing of RBC deformability for salvaged blood (RBC$_{SI}$) and freshly-collected cells (RBC$_{B}$).

| Parameters * | After Salvage | | Before Salvage | | *p* Values |
|---|---|---|---|---|---|
| | Average | ±SD | Average | ±SD | |
| UDFC, % | 3.23 | 1.75 | 2.31 | 1.60 | > 0.05 |

* UDFC—fraction of un-deformable (ER ≤ 1.1) cells in RBC population %.

Furthermore, although Table 1 shows no significant difference in %UDFC between cells before (native) and after salvage, in individual cases the salvage did induce either an increase or decrease in the %UDFC, as shown in Figure 7.

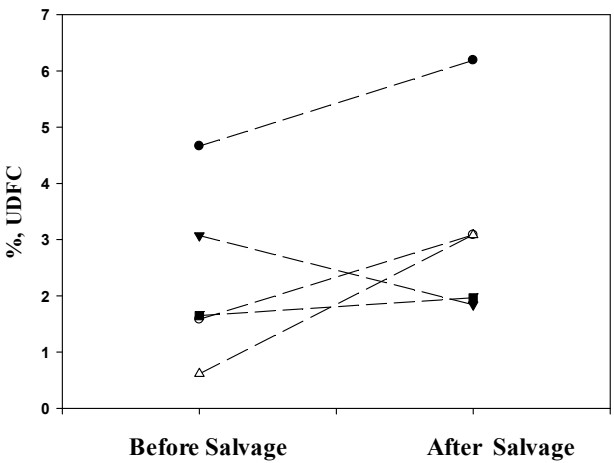

**Figure 7.** Percentage of undeformable cells (%UDFC) in RBC population before and after salvage.

## 4. Discussion

RBC deformability is the ability of cells to adjust their shape to pass through microvessels, especially capillaries, which are narrower than erythrocyte diameter. Increased rigidity interferes with blood perfusion and impairs oxygen delivery to peripheral tissues [48–51]. Rigid RBCs can weaken tissue perfusion and block capillaries [51–53]. Deformability is a function of some basic properties of the erythrocyte, such as membrane surface area-to-volume ratio, cytosol viscosity, and binding stability between the lipid bilayer and cytoskeleton [51,70]. All these factors are affected by the cell environment and external conditions. Additional factors include the shear stress level applied to the RBC and the flow pattern. RBCs are subjected to mechanical stress as they pass through the cardiovascular system but do not sustain significant damage. However, normal erythrocytes can be damaged in non-physiological conditions such as high mechanical stress created by flow in artificial organs or auxiliary devices. This can cause deterioration in the vital properties of RBCs, which reduces their ability to resist further damage and leads to impairment of cell integrity and subsequent hemolysis.

The physiological level of shear stress (up to 5 Pa [71]) enables RBC deformation, detachment of red cells from the endothelium, and disaggregation of rouleaux without inducing immediate or long-term injury to blood components [31]. In contrast, supraphysiological shear stress can impair RBC deformability without initiating hemolysis [31,72].

RBC damage is always observed when blood is exposed to shear stress levels above 5 Pa but below a "hemolytic threshold" [12,31,39,72]. The destruction of RBCs (i.e., hemolysis) can occur when the RBC is exposed to shear stress that exceeds the hemolytic thresh-

old [15,21]. At the same time, it is essential to point out that two factors determine the outcome of a mechanical effect: the level of applied stress and the duration of its application [15].

Although most authors indicate a deterioration in cell deformability following mechanical stress [31,72], some publications report an improvement in deformability [22,73]. A comparison of the various results allows us to conclude that moderate mechanical stress (5–10 Pa) should lead to an improvement in cell deformability [22,73] and high stress should lead to its deterioration [31,72]. Lee et al. [72] showed that RBC deformability begins to decrease when the shear stress exceeds 30 Pa and suggests that deformability measurement can be a marker for showing erythrocyte sub-hemolytic damage.

One of the clinical setups in which mechanical stress is applied to RBCs outside of the body is blood salvage. Intraoperative autologous blood transfusion has expanded in recent decades [9,74]. There are conflicting results regarding the effect of blood salvage on red cell properties [18,75,76]. Vonk et al. [76] report that cell savage "does not severely alter the rheological properties of blood." In line with these results, Salaria et al. [75] demonstrated (for patients undergoing cardiac surgery) that transfusion of salvaged RBCs does not lead to a decrease in deformability and an increase in aggregation of a recipient's red cells.

In contrast, Gu et al. [18] observed that the blood salvage reduced the RBC's deformability but did not affect the RBC aggregation. Thus, we can conclude that there is no unequivocal conclusion in the literature about the effect of mechanical stress on the properties of RBCs. The characteristics of the applied stress will most probably determine the result of mechanical action, but it remains unknown whether the cells' initial (pre-exposure) properties themselves are a factor influencing the observed effect.

In a previous study, we tried to assess the effect of the processing of donated blood into a packed RBC unit (in which the RBCs are subjected to shear stress induced by centrifugation and filtration) on the RBC deformability [58]. We found that the change in the percentage of undeformable cells (%UDFC) was bidirectional, depending on its initial level. The percentage of UDFC increased when its initial level was low and decreased when it was high [58]. In the present study, we set out to determine whether that relationship is specific to the preparation of storage units or whether it can be extended to a broader range of situations in which cells are subjected to moderate mechanical stress. For that purpose, we have analyzed the distribution of RBC deformability following the application of moderate mechanical stress, either in vitro (by rolling RBCs with metal beads) or during blood salvage, focusing on the percentage of undeformable cells, as described in Methods.

For both procedures, in all study groups, we observed, in accord with our previous findings, that the Δ%UDFC was bidirectional, depending on its initial level (Figures 2 and 7); the %UDFC increased when its initial level was low and decreased when it was high.

This phenomenon can be explained by the results presented by Sakota et al. [32] and Yokoyama et al. [77], who demonstrated that in the blood (from Holstein calves or pigs) sheared by rotary pumps, an increase in the MCV of erythrocytes and a decrease in MCHC were observed. The authors [77] speculated that the likely mechanism is that aged erythrocytes with smaller volumes and higher cytosol concentrations of hemoglobin were primarily destroyed, leaving younger cells with higher volumes and lower Hb concentrations. Thus, they assume that the selective destruction of the aged RBCs occurred by removing more fragile cells possessing a lower hemolytic threshold. Due to the aged RBCs' higher fragility [62], i.e., lower resistance to shear stress, their selective destruction leads to an elevation of average MCV and a reduction of MCHC following blood pumping [32,77].

In light of the cited publications [32,77], we can provide the following explanation for our results: Exposure of red cells to moderate mechanical stress initiates a deterioration in the deformability of each cell. However, due to the RBC population being a mixture of cells of different ages and damage level, their exposure to mechanical stress induces differential impairment in cell deformability and selective destruction of "aged"/defective cells. Thus, in our system, we simultaneously observe the results of two processes: the decrease in cell deformability and the "destruction" of non-deformable cells. Therefore, the overall

effect depends on the proportion of undeformable cells in the RBC sample before treatment. With a significant amount of UDFC in the original sample, applying mechanical stress will result in partial hemolysis of these cells and, as a result, a decrease in their proportion in the RBC population. In an inverse situation, an increase in the fraction of undeformable cells is observed.

## 5. Conclusions

The use of intra-operative blood salvage, dialysis, and artificial organs and devices is associated with the application of moderate mechanical stress on RBCs. It has been previously demonstrated that prolonged application of supraphysiological mechanical stress may promote the spread of abnormal erythrocyte morphology [21], phosphatidylserine externalization [40], the elevation of cell fragility [41], and increased vesiculation [40,41]. Ultimately, these changes lead to cell hemolysis [27,31–36].

We analyzed the alteration of rigid RBC subpopulation weight following mechanical stress. The main starting point of our study is the assumption that the RBC population is not homogeneous and, therefore, the response to the mechanical impact depends on the initial state of the cells. We conclude that the overall result depends on the initial proportion of undeformable cells: a stress-induced change in the proportion of rigid cells increased ($\Delta$%UDFC > 0) when its initial value was low, and decreased ($\Delta$%UDFC < 0) when its initial value was high.

The findings and considerations above suggest that moderate mechanical stress induces impairment in RBC deformability. However, since the erythrocyte population is a mixture of cells with different levels of accumulated damage [78,79], the impact of mechanical stress causes a variable response. Thus, cells with a low initial degree of damage partially lose their ability to deform. In contrast, cells with a large number of membrane defects are destroyed (hemolyzed) under the action of the applied mechanical impact. Thus, on the one hand, there is a deterioration in the deformability of most cells, and on the other hand, there is selective destruction of "aged"/inferior cells. Moreover, we have shown that the above conclusions are valid for various cases of mechanical action on RBCs (rotation with steel balls, blood salvage, and preparation of stored RBCs units [58]). This allows us to hope that the results of this study will present the need to consider the patient-to-patient variability of RBC properties when artificial organs are developed.

**Author Contributions:** Conceptualization, S.Y. and G.B.; methodology, G.B.; investigation, T.T. and G.B.; resources, S.B.; writing—original draft preparation, T.T. and G.B.; writing—review and editing, S.Y., S.B. and D.A. and A.G.; supervision, S.Y.; project administration, S.B. All authors have read and agreed to the published version of the manuscript.

**Funding:** This study was supported by a grant from the Israel Science Foundation (to G. Barshtein; 341/18) and from the Haemonetics (Boston, MA, USA) to S. Yedgar and S. Beyth.

**Institutional Review Board Statement:** Blood samples collection from healthy volunteers and stored units after obtaining an Institutional Helsinki Committee Regulations Permit (98290, Hadassah Hospital, Jerusalem, Israel). Operation with blood salvager (OrthoPAT, Haemonetics) realized under permeation of Institutional Review Board (Hadassah Hospital, Jerusalem, Israel).

**Informed Consent Statement:** Informed consent was obtained from all subjects involved in the study.

**Data Availability Statement:** Not applicable.

**Acknowledgments:** We thank Olga Fredman (The Hebrew University, Faculty of Medicine) and B.M. Alyan Muna (Blood Bank, Hadassah University Hospital) for their technical assistance.

**Conflicts of Interest:** The authors declare no conflict of interest.

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
