# Peer review of "Double-Facet Effect of Artificial Mechanical Stress on Red Blood Cell Deformability: Implications for Blood Salvage"

_applsci, doi:10.3390/app12199951_

Round 1
Reviewer 1 Report
1) Line 80: Please provide the full citation for your previous study. Please clearly state how the present study differs from your previous study.
2) Line 81: Please define what you mean by the “initial state” of the donor cells? Initial state in what sense? Age?
3) Line 120: Verb is missing.
4) I recommend providing the centrifugation unit in “g” (force units) instead of rpm for proper standardization and making it independent of the centrifugation instrument.
5) Why were outdated packed RBC units used? They could very well have altered biomechanical and altered biochemical characteristics (which in turn could lead to altered biomechanical characteristics). This could have a major effect on the results shown in Figure 1. If this was intentional, this should be stated upfront (in the methods section).
6) Please provide the magnitude of the shear stress induced by the rolling bead method. Please provide a bit more detailed description of this method since it does not seem to be a very common method used, and some readers, including myself are not familiar with this method. In the discussion, you talk about stress levels, but you don’t report the stress values that you have applied.
7) Why was albumin and why 0.5% was used in deformability measurements?
8) Please indicate if a one-tailed or a two-tailed statistical test was used.
9) Please provide the actual value of “light mechanical stress.”
10) How do you know that hemolysis of less than 5% is solely due to the applied mechanical stress, and not induced earlier by other factors (e.g., biochemical stress, centrifugation, etc.)?
11) Please provide a mechanistic explanation for the results shown in Figure 2.
12) There are some marks below Figure 3.
13) There should be a scale for Figure 5A.
14) Sphericity index is a more elegant quantity than ER since it accounts for volume and surface area. Authors may wish to consider using this parameter.
15) Most of the native as well as salvaged cells shown in Figure 5A more or less exhibit the normal morphology of RBCs (discoidal), consistent with results shown in Figure 5B. So what is the significance of this and what does it say about the salvage procedure? What is the new significant finding here?
16) In line 142, you mention “light mechanical stress,” yet in line 376 you indicate “moderate mechanical stress.” There seems to be an inconsistency. Reporting your actual stress values would make it quantitative and comparable the work by others in the field.
17) Some of the conclusions and consistencies in the literature about the effects of salvage could be due to the methods used by various investigators for quantification of the mechanics.
Author Response
We thank the reviewer for taking the trouble to read our article and giving us useful recommendations for improving the text. We hope that our responses and the manuscript changes comply with the reviewer's guidance.
1. Line 80: Please provide the full citation for your previous study. Please clearly state how the present study differs from your previous study.
We thank the reviewer for the comment; the text was modified accordingly (see Lines 131-143, the revised version of the manuscript).
2. Line 81: Please define what you mean by the “initial state” of the donor cells? Initial state in what sense? Age?
We thank the reviewer for the comment; the text was modified accordingly (see Lines 144 & 582, the new version of the manuscript).
3) Line 120: Verb is missing.
We thank the reviewer for the comment; the text was modified accordingly (see Line 227, the revised version of the manuscript).
4. I recommend providing the centrifugation unit in “g” (force units) instead of rpm for proper standardization and making it independent of the centrifugation instrument.
We thank the reviewer for the comment; the text was modified accordingly (see Line 230, the revised version of the manuscript).
5. Why were outdated packed RBC units used? They could very well have altered biomechanical and biochemical characteristics (which could lead to altered biomechanical characteristics). This could have a significant effect on the results shown in Figure 1. This should be stated upfront (in the methods section) if this was intentional.
We agree with the reviewer that the storage of cells leads to a significant change in their properties. However, the drawing of RBCs from a unit before the end of its clinical use is very problematic due to ethical issues. For this reason, as written in line 126 (original document), we tested samples whose clinical use had expired no more than three days before sampling. So, we believe that the behavior of tested RBCs adequately reflects the properties of cells at the later stages of storage, which are characterized by a high concentration of undeformed cells.
Following the reviewer's recommendation, we have included relevant clarifications in the revised text of the manuscript (Lines 234 & 235).
6. Please provide the magnitude of the shear stress induced by the rolling bead method. Please provide a bit more detailed description of this method since it does not seem to be a very common method used, and some readers, including myself, are not familiar with this method. In the discussion, you talk about stress levels but don't report the stress values you have applied.
Unfortunately, we cannot estimate the shear stress level applied to the cells during used mechanical stress method. However, we would like to draw attention to the fact that this method is classical in studying the mechanical fragility of cells. We added the corresponding comment to the text of the revised document (see Line 247).
7. Why was albumin and why 0.5% was used in deformability measurements?
Albumin has been routinely used for keeping red cells in shape. In our experiments, we used 0.5% albumin in deformability measurements due to a higher protein concentration, thereby provoking RBCs detachment from the slide.
8. Please indicate if a one-tailed or a two-tailed statistical test was used.
We used a two-tailed test (see Line 340, the revised manuscript version).
9. Please provide the actual value of “light mechanical stress.”
As described previously, we cannot characterize the level of applied stress. For each sample, conditions of the stress application are equal, and the hemolysis level for each test is less than 5.0 %, as demonstrated by us previously [57]. Please see Line 340 in the revised version of the manuscript.
10) How do you know that less than 5% hemolysis is solely due to the applied mechanical stress and not induced earlier by other factors (e.g., biochemical stress, centrifugation, etc.)?
Before each application of shear stress, we washed the cells, and the level of Hb in the supernatant was negligible.
11) Please provide a mechanistic explanation for the results shown in Figure 2.
We thank the reviewer for the question. An appropriate comment has been added to the revised text (see Lines 371-376, the revised version of the manuscript).
12) There are some marks below Figure 3.
We thank the reviewer for the comment; the text was modified accordingly.
13) There should be a scale for Figure 5A.
We thank the reviewer for the comment; the text was modified accordingly (see Line 452, the revised version of the manuscript).
14) Sphericity index is a more elegant quantity than ER since it accounts for volume and surface area. Authors may wish to consider using this parameter.
We thank the reviewer for the interesting suggestion. However, we have used the index for over 20 years (in dozens of publications) and found it adequate. In addition, this parameter is also used in the publications of many other groups. For this reason, we would not like to violate the already accepted system of notation.
15) Most of the native as well as salvaged cells shown in Figure 5A more or less exhibit the normal morphology of RBCs (discoidal), consistent with results shown in Figure 5B. So what is the significance of this and what does it say about the salvage procedure? What is the new significant finding here?
As stated in our manuscript for mechanical stress (created by rolling steel balls) and blood salvage, we do not observe changes when statistically comparing the number of undeformed cells before and after stress application. However, a paired comparison of the results leads to the conclusion that both an increase and a decrease in the discussed indicator are possible.
Thus, in combination with our previous results [54], we demonstrated that our conclusions are valid for the model system (rolling balls) and clinically relevant processes. We have added the corresponding phrase to the text of the edited manuscript (see Lines 596-620, the revised version of the manuscript).
16) In line 142, you mention “light mechanical stress,” yet in line 376 you indicate “moderate mechanical stress.” There seems to be an inconsistency. Reporting your actual stress values would make it quantitative and comparable to the work of others in the field.
We thank the reviewer for the comment. As we pointed out earlier, we cannot calculate the applied shear stress. We have made the necessary changes in the corrected text of the manuscript (see Lines 251 and 346, the revised version of the manuscript).
17) Some of the conclusions and consistencies in the literature about the effects of salvage could be due to the methods used by various investigators to quantify the mechanics.
We fully agree with the reviewer's comment (see Lines 35-39, the revised version of the manuscript).

Reviewer 2 Report
Double-Facet Effect of Artificial Shear Stress on Red Blood Cell Deformability: Implications for Blood Salvage
The authors have studied the effect of blood saving procedures on the deformability of the RBCs. The mechanical action of stress affects the membrane fluidity.
As calcium is one of the cellular molecules that contribute to the hardening of RBCs. It will be interesting to show the difference in calcium content of the rigid undeformable cells in comparison to normal RBCs
The manuscript can be accepted with the revision.
Specific comments
Abstract: Line 19 In vitro, italics
Figure 1. Add the details for the symbols used
Figure 5. Correct the spelling for Ratio. The native and salvaged elongation ratio can be represented separately
Author Response
We thank the reviewer for taking the trouble to read our article and giving us useful recommendations for improving the text. We hope that our responses and the manuscript changes comply with the reviewer's guidance.
Reviewer:
The authors have studied the effect of blood saving procedures on the deformability of the RBCs. The mechanical action of stress affects the membrane fluidity.
As calcium is one of the cellular molecules that contribute to the hardening of RBCs. It will be interesting to show the difference in calcium content of the rigid undeformable cells in comparison to normal RBCs
We thank the reviewer for the comment; however, establishing the mechanism of action of mechanical stress on the state of the cytosol and cell membrane was not the goal of our study. We want to draw the reviewer's attention to the publication of Liao et al. [28], which discusses the role of calcium in erythrocyte damage during blood rescue.
The manuscript can be accepted with the revision.
Specific comments
- Abstract: Line 19 In vitro, italics
We thank the reviewer for the comment; the text was modified accordingly (see Line 18, the revised version of the manuscript).
- Figure 1. Add the details for the symbols used
We thank the reviewer for the comment; the text was modified accordingly (see Lines 365-367, the revised version of the manuscript).
- Figure 5. Correct the spelling for Ratio. The native and salvaged elongation ratio can be represented separately.
Here we are forced to disagree with the reviewer. We showed the averaged curves for two types of samples (before and after blood salvage) on the same figure so that you can see that they practically coincide. If we present them separately, it will be impossible to make the corresponding conclusion visually.
Reviewer 3 Report
The authors investigated the effect of shear stress on the deformability of RBCs in their paper titled "Double-Facet Effect of Artificial Shear Stress on Red Blood Cell Deformability: Implications for Blood Salvage". The manuscript is well-written, and I found it interesting. However, minor revisions are required. The manuscript should be revised in response to the comments.
1. The authors should seek editing assistance from someone with full professional proficiency in English.
2. At the end of the "abstract," a concluding statement is required.
3. The "abstract" is not structured. It should be written in a single paragraph.
4. In the "introduction," the authors should elaborate on the effects of artificial organs and devices on RBCs.
5. The catalogue numbers of the materials used in this study should be included.
6. Changes that are statistically significant should be noted in the captions of the figures.
7. The "Conclusion" section is poorly written. This section should begin with a background statement. At the end, a concluding statement is also required.
8. The manuscript should be prepared in accordance with the "Guidelines for authors".
9. Figure 5 (A) should have scale bars.
10. Citations are required for some of the sentences throughout the manuscript. For example:
- On page 1, the first paragraph of the "Introduction", "Significant blood loss during surgery may be … has a negative effect on organ vitality."
- On pages 8 and 9, lines 325-335, "Deformability is a function of some basic properties … leads to impairment of cell 334 integrity and subsequent hemolysis."
11. All "in vitro", "ex-vivo", and "et al." should be written in italics throughout the manuscript.
Author Response
We thank the reviewer for taking the trouble to read our article and giving us useful recommendations for improving the text. We hope our responses and the manuscript changes comply with the reviewer's guidance.
Reviewer
The authors investigated the effect of shear stress on the deformability of RBCs in their paper titled "Double-Facet Effect of Artificial Shear Stress on Red Blood Cell Deformability: Implications for Blood Salvage". The manuscript is well-written, and I found it interesting. However, minor revisions are required. The manuscript should be revised in response to the comments.
- The authors should seek editing assistance from someone with full professional proficiency in English.
We thank the reviewer for the comment; the text was modified accordingly.
- At the end of the "abstract," a concluding statement is required.
We thank the reviewer for the comment; the text was modified accordingly; see Lines 27-28, the revised version of the manuscript.
- The "abstract" is not structured. It should be written in a single paragraph.
We thank the reviewer for the comment; the text was modified accordingly.
- In the "introduction," the authors should elaborate on the effects of artificial organs and devices on RBCs.
We thank the reviewer for the comment; the text was modified accordingly; see Lines 80-98, the revised version of the manuscript.
- The catalogue numbers of the materials used in this study should be included.
We thank the reviewer for the comment; the text was modified accordingly; see Lines 213-217 (revised version of the manuscript).
- Changes that are statistically significant should be noted in the captions of the figures.
We thank the reviewer for the comment; the text was modified accordingly; see Lines 367, 402, 420, and 443 (revised version of the manuscript).
- The "Conclusion" section is poorly written. This section should begin with a background statement. At the end, a concluding statement is also required.
We thank the reviewer for the comment; the text was modified accordingly; see Lines 598-606 & 618-622 (revised version of the manuscript).
- The manuscript should be prepared in accordance with the "Guidelines for authors".
We thank the reviewer for the comment; the text was modified accordingly.
- Figure 5 (A) should have scale bars.
We thank the reviewer for the comment; the text was modified accordingly; see Line 454 (revised version of the manuscript).
- Citations are required for some of the sentences throughout the manuscript. For example:
- On page 1, the first paragraph of the "Introduction", "Significant blood loss during surgery may be … has a negative effect on organ vitality."
- On pages 8 and 9, lines 325-335, "Deformability is a function of some basic properties … leads to impairment of cell 334 integrity and subsequent hemolysis."
We thank the reviewer for the comment; the text was modified accordingly; see Line 32 and 516 (revised version of the manuscript).
- All "in vitro", "ex-vivo", and "et al." should be written in italics throughout the manuscript.
We thank the reviewer for the comment; the text was modified accordingly.
Reviewer 4 Report
In this article, the author reported the double-sided effect of artificial shear stress on the deformability of red blood cells and measured the deformability of red blood cells before and after the application of mechanical stress through in vitro systems and blood-saving procedures used in clinical practice. However, some aspects deserve careful study and consideration. I have a few suggestions for your reference.
1. Introduction the indications, advantages, and disadvantages of blood recovery for clinical patients need to be correctly mentioned in the background introduction, which indicates the importance of this study for blood recovery in clinical practice.
2. The best experimental method and procedure is to make a flow chart, which should make the reader more concise and clear.
3. Results: there were no deformed cells in the red blood cell population of fresh and stored samples( Δ Udfc,%) and the percentage of undeformed cells in RBC population after increasing mechanical stress( Δ Udfc,%) is too simple, and it is recommended to remap.
4. The results of the two shear stress application methods mentioned in the paper on the deformation ability of red blood cells are not compared, and it is unclear which of the two methods has a minor impact on blood recovery.
5. It was mentioned in the discussion that the red blood cell population is a mixture of cells of different ages and sustained damage. Still, the age and population of red blood cells extracted in the design of experimental methods are not precise. How to compare and explain?
6. Conclusion: it is concluded that mechanical stress permanently damages red blood cells. Still, no method is proposed to minimize the damage of such mechanical stress to red blood cells, which should be considered and prospected.
Author Response
We thank the reviewer for taking the trouble to read our article and giving us useful recommendations for improving the text. We hope our responses and the manuscript changes comply with the reviewer's guidance.
Reviewer:
In this article, the author reported the double-sided effect of artificial shear stress on the deformability of red blood cells and measured the deformability of red blood cells before and after the application of mechanical stress through in vitro systems and blood-saving procedures used in clinical practice. However, some aspects deserve careful study and consideration. I have a few suggestions for your reference.
- Introduction the indications, advantages, and disadvantages of blood recovery for clinical patients need to be correctly mentioned in the background introduction, which indicates the importance of this study for blood recovery in clinical practice.
We thank the reviewer for the comment; the text was modified accordingly; see Lines 102-110 (revised version of the manuscript).
- The best experimental method and procedure is to make a flow chart, which should make the reader more concise and clear.
We thank the reviewer for the comment; the text was modified accordingly, see Figure 1 (revised version of the manuscript).
- Results: there were no deformed cells in the red blood cell population of fresh and stored samples( Δ Udfc,%) and the percentage of undeformed cells in RBC population after increasing mechanical stress( Δ Udfc,%) is too simple, and it is recommended to remap.
We thank the reviewer for the comment; the text was modified accordingly. The relevant paragraph has been added to the text of the revised version of the manuscript (Lines 385 - 390).
- The results of the two shear stress application methods mentioned in the paper on the deformation ability of red blood cells are not compared, and it is unclear which of the two methods has a minor impact on blood recovery.
We thank the reviewer for asking the question. The relevant paragraph has been added to the text of the revised version of the manuscript; see Lines 628-634 (revised version of the manuscript).
- It was mentioned in the discussion that the red blood cell population is a mixture of cells of different ages and sustained damage. Still, the age and population of red blood cells extracted in the design of experimental methods are not precise. How to compare and explain?
We thank the reviewer for asking the question. The relevant paragraph has been added to the text of the revised version of the manuscript; see Lines 617-619.
- Conclusion: it is concluded that mechanical stress permanently damages red blood cells. Still, no method is proposed to minimize the damage of such mechanical stress to red blood cells, which should be considered and prospected.
We thank the reviewer for asking the question. The relevant paragraph has been added to the text of the revised version of the manuscript; see Lines 630-634.
Round 2
Reviewer 1 Report
Authors have more or less addressed the comments.
Author Response
I would like to extend our sincere thanks to reviewer for the constructive comments and helpful suggestions on our manuscript.
Reviewer 2 Report
Double-Facet Effect of Artificial Shear Stress on Red Blood Cell Deformability: Implications for Blood Salvage
The authors have studied the effect of blood saving procedures on the deformability of the RBCs. The mechanical action of stress affects the membrane fluidity.
As calcium is one of the cellular molecules that contribute to the hardening of RBCs. It will be interesting to show the difference in calcium content of the rigid undeformable cells in comparison to normal RBCs
The manuscript can be accepted with the revision.
Specific comment:
Figure 5. Correct the spelling for Ratio. The native and salvaged elongation ratio can be represented separately
Author Response
I would like to extend our sincere thanks to reviewer for the constructive comments and helpful suggestions on our manuscript.
Following the reviewer's recommendation, we have modified Figure 6 (Figure 5 in the original version) and made appropriate changes to the figure legend.
Reviewer 4 Report
This manuscript has addressed previous concerns and can be published in its current form.
Author Response
I sincerely thank the reviewer for the constructive comments and helpful suggestions on our manuscript.